# Learning Density Distribution of Reachable States for Autonomous Systems

**Yue Meng**
MIT
United States
mengyue@mit.edu

**Dawei Sun**
UIUC
United States
daweis2@illinois.edu

**Zeng Qiu**
Ford
United States
cqiu1@ford.com

**Md Tawhid Bin Waez**
Ford
United States
mwaez@ford.com

**Chuchu Fan**
MIT
United States
chuchu@mit.edu

**Abstract:** State density distribution, in contrast to worst-case reachability, can be leveraged for safety-related problems to better quantify the likelihood of the risk for potentially hazardous situations. In this work, we propose a data-driven method to compute the density distribution of reachable states for nonlinear and even black-box systems. Our semi-supervised approach learns system dynamics and the state density jointly from trajectory data, guided by the fact that the state density evolution follows the Liouville partial differential equation. With the help of neural network reachability tools, our approach can estimate the set of all possible future states as well as their density. Moreover, we could perform online safety verification with probability ranges for unsafe behaviors to occur. We use an extensive set of experiments to show that our learned solution can produce a much more accurate estimate on density distribution, and can quantify risks less conservatively and flexibly comparing with worst-case analysis.

**Keywords:** Reachability Density, Machine Learning, Liouville Theorem

## 1 Introduction

Reachability analysis has been a central topic and the key for the verification of safety-critical autonomous systems. The majority of existing reachability approaches either compute worst-case reachable sets [1, 2, 3, 4, 5, 6], or use Monte Carlo simulation to estimate the reachable states with probabilistic guarantees [7, 8, 9, 2]. There are several obvious disadvantages

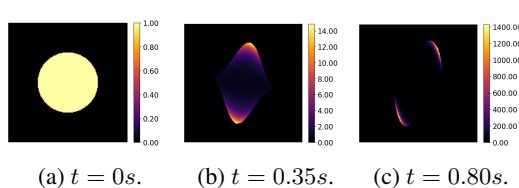

(a) $t = 0s$.  (b) $t = 0.35s$.  (c) $t = 0.80s$.
Figure 1: Reachability density of Van der Pol.

of such methods. 1). Worst-case methods, especially those for nonlinear systems, often over-approximate the reachable sets and produce very conservative results (due to convex set representations [10, 11, 12, 13], wrap effects [14, 15], low-order approximation [3, 16], etc.), not to say they are usually very computationally expensive (also known as the curse of dimensionality). 2). Existing methods do not care about reachable state concentration and produce the same reachable state estimations for different distributions of initial states if those distributions share the same support [1]. That is, current approaches do not compute which states are more likely to be reached. For example, in Fig. 1, the state density distribution of a Van der Pol oscillator evolves over time and concentrates on certain states (the highlighted part in the figure), even if the initial states are uniformly distributed. If one uses an existing worst-case reachability algorithm, most likely the results in Fig. 1 (b)(c) will show almost the entire black region inside the highlighted part is reachable, as those methods often use convex sets to represent the reachable sets.

---

[1]The probabilistic guarantees of the sampling-based methods do rely on the form of the initial states distribution. However, the final reachable sets estimate is the same for different distributions with the same support.

5th Conference on Robot Learning (CoRL 2021), London, UK.

In this paper, we aim to tackle the above problems and propose a learning-based method that can compute the density distribution of the reachable states from given initial distributions. The state density is much more powerful than worst-case reachability and can better quantify risks. Our proposed method is based on the Liouville theorem [17, 18, 19], which is from classical Hamiltonian mechanics and asserts that the state density distribution function (which is the measurement of state concentration) is constant along the trajectories of the system. Given an autonomous system $\dot{x} = f(x)$ that is locally Lipschitz continuous, the evolution of the state density $\rho(x,t)$ (i.e. the density of state $x$ at time $t$) is characterized by a Liouville partial differential equation (PDE). We learn the state density as a neural network (NN) while respecting the laws of physics described by the nonlinear Liouville PDE, in a way that is similar to the physics-informed NN [20].

Furthermore, we make two major improvements to make the learned density NN suitable for verification. 1). Instead of learning $\rho$ for a fixed initial distribution $\rho(x_0, 0)$, we learn the density concentration function which specifies the multiplicative change of density from any $\rho(x_0, 0)$. 2). We use a single NN to jointly learn both the reachable state $\Phi(x_0, t)$ and its corresponding density. Moreover, we use Reachable Polyhedral Marching (RPM) [21]—an exact ReLU NN reachability tool—to parse our learned NN as linear mappings from input polyhedrons to output polyhedrons. Using such parsed polyhedrons, we can perform online forward and backward reachability analysis and get the range of density bounds $[\rho_{\min}, \rho_{\max}]$ for each output polyhedron. Together, our method can perform online safety verification by computing the probability of safety (instead of a single yes or no answer) under various initial conditions, even with unknown system dynamics $f$ (i.e. a black-box system where we only have access to a simulator of $x(t)$). In this way, it also has the potential to collect environmental data in run-time and update its distribution for online safety verification.

We conduct experiments on 10 different benchmarks covering systems from low dimension academic examples to high dimension black-box simulators equipped with either hand-crafted or NN controllers. Surprisingly, without using any ground-truth density data in the learning process, our approach can achieve up to 99.89% of error reduction in KL divergence with respect to the ground-truth value, when compared to sampling-based methods like kernel density estimation and Gaussian Processes. Moreover, our learned density concentration function can also be used for reachability distribution analysis. We show that in several systems, more than 90% of the states can actually just reside in a small region (less than 10% of the volume of the convex hull for those states) in the state space, which also points out the conservativeness of the worst-case reachability analysis methods in terms of quantifying risks. We also show that different initial distributions can lead to a drastic change in the safety probability, which can help in cases when unsafe is inevitable but can be designed to happen with a very low probability.

Our major contributions are: (1) we are the first to provide an explicit probability density function for reachable states of dynamical systems characterized as ordinary differential equations; this density function can be used for online safety analysis, (2) we propose the first data-driven method to learn the state density evolution and give accurate state density estimation for arbitrary (bounded support) initial density conditions without retraining or fine-tuning, and (3) we use a variety of examples to show the necessity to perform reachability distribution analysis instead of pure worst-case reachability, to flexibly and less conservatively quantify safety risks.

**Related work** Reachability analysis, especially worst-case reachability using sampling-based methods or set propagation, has been studied for decades. The literature on reachability has been extensively studied in many surveys [1, 6, 22, 23]. Here we only discuss a few closely related works.

Hamilton Jacobian PDE has been used to derive the exact reachable sets in [24, 1, 5]. However, the HJ-PDE does not provide the density information. Many data-driven approaches can compute probabilistic reachable sets using scenario optimization [7, 25], convex shapes [8, 9, 26], support vector machines [27, 28], kernel embedding [29], active learning [30], and Gaussian process [2]. However, the probabilistic guarantees they provide are usually in the form that $\mathbf{Prob}(x(t) \in \text{estimated set}) > 1 - \epsilon$ with enough samples, instead of the state density distribution. [31] estimates human state distribution using a probabilistic model but requires state discretization. [32] uses the Liouville equation to maximize the backward reachable set for only polynomial system dynamics. In [33] the authors compute the stochastic reachability by discretizing stochastic hybrid systems to Markov Chains (MC), then perform probabilistic analysis on the discretized MC. The closed-form expression of the probability requires integrals over the whole state space hence is computation-heavy and cannot be used for online safety check. The closest to ours

is [34] where Perron-Frobenius and Koopman operators are learned from samples of trajectories. Then, the learned operators can be used to transform the moments of the distribution over time. The distribution at time $t$ is then recovered from the transformed moments. However, as they use moments up to a specific order to represent a distribution, even if the learned operators are perfect, the estimation error of the distribution might not be zero. Also, such a moment-based method is hard to scale to large-dimensional systems. Recently, there is also a growing interest in studying the (worst-case) reachability of NN [35, 36, 37, 38, 21] or systems with NN controllers [39, 4, 40, 41, 42]. In this paper, we use [21] to parse our learned NN as a set of linear mappings between polyhedrons for online reachability computation, but this step can be replaced with other NN reachability tools.

To measure the probability distribution in the reachable sets, the most naïve approach is to use histograms or kernel density estimation [19], similar to the Monte-Carlo method used in dispersion analysis [43]. But this could lead to poor accuracy and computational scalability [44]. Another approach propagates the uncertainty by approximating the solution of the probability density function (PDF) transport equation [45], which could still be time-consuming due to the optimization process performed at each time step. Our approach finds the PDF transport equation by solving the Liouville PDE [2] using NN. Similar ideas have been explored in [18]. However, the density NN in [18] was learned solely for a fixed initial states distribution and therefore, cannot be used for online prediction. Instead, we jointly learn the reachable states and density changes, and can perform online reachability density computation for any initial states distribution.

The idea of using deep learning to solve PDEs can be traced back to the 1990s [46, 47, 48], where the solutions of the PDE on a priori fixed mesh are approximated by NN. Recently, there is a growing interest in the related sub-fields including: mesh-free methods in strong form [20, 49, 50] and weak form [51, 52], solving high-dimension PDEs via BSDE method [53], solving parametric PDE [54, 55] and stochastic differential equations [56, 57], learning differential operators[58, 59, 60], and developing more advanced toolboxes [61, 62, 63]. Our idea for solving Liouville PDE along trajectories is similar to [53] but without the stochastic term.

## 2  Preliminaries

We denote by $\mathbb{R}, \mathbb{R}^{\geq 0}, \mathbb{R}^n, \mathbb{R}^{n \times n}$ the sets of real numbers, non-negative real numbers, $n$-dimensional real vectors, and $n \times n$ real matrices. We consider autonomous dynamical systems of the form $\dot{x}(t) = f(x(t))$, where for all $t \in \mathbb{R}$, $x(t) \in \mathcal{X} \subseteq \mathbb{R}^d$ is the state and $\mathcal{X}$ is a compact set that represents the state space. We assume that $f : \mathbb{R}^d \mapsto \mathbb{R}^d$ is locally Lipschitz continuous. The solution of the above differential equation exists and is unique for a given initial condition $x_0$. We define the flow map $\Phi : \mathcal{X} \times \mathbb{R} \mapsto \mathcal{X}$ such that $\Phi(x_0, t)$ (also written as $x(t)$ for brevity) is the state at time $t$ starting from $x_0$ at time $0$. Note that system parameters can be easily incorporated as additional state variables with time derivative to be $0$.

We analyze the evolution of the dynamical system by equipping it with a density function $\rho : \mathcal{X} \times \mathbb{R} \to \mathbb{R}^{\geq 0}$, which measures how states distribute in the state space at a specific time instant. A larger density $\rho(x, t)$ means the state is more likely to reside around $x$ at time $t$, and vice versa. The density function is completely determined by the underlying dynamics (i.e., function $f$) and the initial density map $\rho_0 : \mathcal{X} \mapsto \mathbb{R}^{\geq 0}$. Specifically, given a $\rho_0$, the density function $\rho$ solves the following boundary value problem of the Liouville PDE [17]:

$$\frac{\partial \rho}{\partial t} + \nabla \cdot (\rho \cdot f) = 0, \quad \rho(x, 0) = \rho_0(x), \tag{1}$$

where $\nabla \cdot (\rho \cdot f) = \sum_{i=1}^d \frac{\partial [\rho \cdot f]_i}{\partial [x]_i}$ is the divergence for the vector field $\rho \cdot f$, and $[\cdot]_i$ takes the $i$-th coordinate of a vector.[3] Intuitively, as shown in [17], Liouville PDE is analogous to the mass conservation in fluid mechanics, where the change of density $\frac{\partial \rho}{\partial t}$ at one point is balanced by the total flux traversing the surface of a small volume surrounding that point. It is hard to solve the Liouville PDE for a closed from of the density function $\rho$. However, it is relatively easy to evaluate

---

[2]In the absence of process noise, this PDF transport equation reduces to stochastic Liouville equation [17]

[3]A general form of Liouville PDE can have a non-zero term on the right-hand side indicating how many (new) states appear or exit from the system during the run time [19]. In all the systems we discuss here, there is no state entering (other than the initial states) or leaving the system, so the right-hand side of Eq. (1) is zero. The total density of the systems we consider is invariant over time.

the density along a trajectory of the system. To do that, we first convert the PDE into an ODE as follows. Considering a trajectory $\Phi(x_0, t)$, the density along this trajectory is an univariate function of $t$, i.e., $\rho(t) = \rho(\Phi(x_0, t), t)$. If we consider the augmented system with states $[x, \rho]$, from Eq. (1) we can easily get the dynamics of the augmented system [19]:

$$\begin{bmatrix} \dot{x} \\ \dot{\rho} \end{bmatrix} = \begin{bmatrix} f(x) \\ -\nabla \cdot f(x)\rho \end{bmatrix}. \tag{2}$$

Therefore, to compute the density at an arbitrary point $x_T \in \mathcal{X}$ at time $T$, one can simply proceed as follows: First, find the initial state $x_0 = \Phi(x_T, -T)$ using the inverse dynamics $-f$. Then, solve the Eq. (2) with initial condition $[x_0, \rho_0(x_0)]$. The solution at time $T$ just gives the desired density value. However, such a procedure only gives the density at a single point, therefore, cannot be used in reachability analysis. Instead, we need to compute the solution of Eq. (2) for a set of initial conditions and use that to compute the reachable sets. To achieve this, we use an NN with ReLU (Rectified Linear Unit) activation functions [64] to jointly approximate the flow map $\Phi$ and the *density concentration function*, as will be shown in the next section.

## 3 Density learning and online reachability density computation

Let us take a closer look at Eq. (2). For an initial condition $[x_0, \rho_0(x_0)]$, the closed form of the solution $\rho$ is $\rho(\Phi(x_0, t), t) = \rho_0(x_0) \exp\left(-\int_0^t \nabla \cdot f(\Phi(x_0, \tau))d\tau\right)$. Interestingly, the solution of $\rho$ is linear in the initial condition $\rho_0$. The gained part denoted by $G(x_0, t) := \exp\left(-\int_0^t \nabla \cdot f(\Phi(x_0, \tau))d\tau\right)$ is a function of the initial state $x_0$ and time $t$, but is independent of $\rho_0$. This mapping $G$ is completely determined by the underlying dynamics and we call it the *density concentration function*. With $G$ in hand, the density at an arbitrary time and state can be quickly computed from any $\rho_0$. However, $G$ is obviously hard to compute. Therefore, we use NN to approximate the *density concentration function*. In addition to $G$, we also use NN to learn the flow map $\Phi$, which will be a necessity for computing the reachable set distribution shown in Sec. 3.2.

### 3.1 System dynamics and density learning framework

Let the parameterized versions of the flow map and the *density concentration function* be $\Phi_\omega$ and $G_\theta$ respectively, where $\omega$ and $\theta$ are parameters. To train the neural network, we construct a dataset by randomly sampling $N$ trajectories of the system: $\mathcal{D}_{train} = \{\xi_i\}_{i=0}^{N-1}$ in $T$ time steps (with time interval $\Delta t$): $\xi_i = \{(x_0^i, 0), (x_1^i, \Delta t), ..., (x_{T-1}^i, (T-1)\Delta t)\}$ where $x_j^i = \Phi(x_0^i, j\Delta t)$. Then, the goal of the learning is to find parameters $\omega$ and $\theta$ satisfying

$$\begin{cases} \Phi_\omega(x_0^i, k\Delta t) - x_k^i = 0, \ \forall i, k, \\ \frac{\partial G_\theta(x_0^i, k\Delta t)}{\partial t} + G_\theta(x_0^i, k\Delta t) \cdot (\nabla \cdot f(x_k^i)) = 0, \ \forall i, k, \end{cases} \tag{3}$$

where the first constraint is for the flow map estimation, and the second constraint enforces the Liouville equation for all the data points.

As for the implementation, we model $\Phi_\omega$ and $G_\theta$ jointly as a fully-connected neural network $\text{NN}(\cdot, \cdot)$ with ReLU activations. To ensure numerical stability as $G$ is an exponential function, we add a nonlinear transform from the NN output to the *density concentration function*:

$$\begin{cases} G_\theta(x_0, t) = \exp(t \cdot \text{NN}_{[0:1]}(x_0, t)) = \exp(t \cdot z(x_0, t)), \\ \Phi_\omega(x_0, t) = \text{NN}_{[1:n+1]}(x_0, t), \end{cases} \tag{4}$$

where $\text{NN}_{[i:j+1]}$ is to choose the $i, i+1, ..., j$-th dimensions from the output of the NN, and $z$ is the intermediate density estimation from the NN. In this way, we guarantee that the *density concentration function* at $t = 0$ is always 1. We optimize our NN via back propagation with the loss function:

$$\mathcal{L} = \lambda \cdot \sum_{i,k} \left[\Phi_\omega(x_0^i, k\Delta t) - x_k^i\right]^2 + \sum_{i,k} \left[\dot{G}_\theta(x_0^i, k\Delta t) + G_\theta(x_0^i, k\Delta t)\left(\nabla \cdot f(x_k^i)\right)\right]^2, \tag{5}$$

where the first term denotes the state estimator square error [65], the second term indicates how far (in the sense of L2-norm) the solution deviates from the Liouville Equation, and $\lambda$ balances

these two loss terms. We approximate the time derivative of the *density concentration function* by $\dot{G}_\theta(x_0^i, k\Delta t) = \left[G_\theta(x_0^i, (k+1)\Delta t) - G_\theta(x_0^i, k\Delta t)\right]/\Delta t$. Our method can also work for black-box systems if we approximate $\nabla \cdot f(\cdot)$ numerically. With some tools from statistical learning theory, we can show that with a large enough number $N$ of samples, the learned flow map and the *density concentration function* can be arbitrarily accurate. A formal proof is shown in Appendix A.

### 3.2  System reachable set distribution computation via NN Reachability Analaysis

In the last section, we have learned an NN that can estimate the density at a single point given the initial density. In this section, we will boost this single-point estimation to set-based estimation by analyzing the reachability of the learned NN. To compute the set of all reachable states and the corresponding density from a set of initial conditions, we use the Reachable Polyhedron Marching (RPM) method [21] to further process the learned NN for $G$ and $\Phi$. RPM is a polyhedron-based approach for exact reachability analysis for ReLU NNs. It partitions the input space into polyhedral cells so that in each cell the ReLU activation map does not change and the NN becomes a fixed affine mapping. With the cells and the corresponding affine mapping on each cell, the exact reachable set for a set of input can be quickly evaluated. Backward reachability analysis can also be performed by computing the intersection of the pre-image of a query output set with the input polyhedron cells.

**System forward reachable set with density.** Recall that the input of the NN in Sec. 3.1 consists of the initial state $x_0$ and time $t$. For the simplicity of comparison with other methods, we only estimate the reachable set and the density at given fixed time instances $t$ and fix the last element of the input of the NN. Thus for each time step t, the input polyhedral cells generated from the RPM will be a set of linear inequality constraints on $x$, and those input cells together with the set of the affine mappings and output polyhedral cells can be represented as: $\{(A_k, b_k, C_k, d_k, E_k, f_k)\}_{k=1}^N$, where in each input cell $H_k := \{v \in \mathbb{R}^d | A_k v \leq b_k\}$, the NN becomes an affine mapping $y = C_k v + d_k$, and thus the image of the input cell is also a polyhedron $M_k = \{y \in \mathbb{R}^{d+1} | E_k y \leq f_k\}$.

Recall that the first dimension of our NN output estimates the *density concentration function* $z$, and the rest dimensions estimate the state $x$, thus the output cell can be written as $M_k = \{(z, x)|E_k[z, x]^T \leq f_k\}$. By projecting it to the state space, we get the reachable set of the system, i.e., $R_k^o := \{x \in \mathcal{X} | (z, x) \in M_k\}$. Then, in each cell $M_k$, we evaluate the lower and upper bounds of $z$, and denote them by $z_{k,min}$ and $z_{k,max}$. The density bound for cell $M_k$ is then computed as

$$\begin{cases} \rho_{k,min} = \rho_0(\bar{x}_k) \cdot \exp(t \cdot z_{k,min}), \\ \rho_{k,max} = \rho_0(\bar{x}_k) \cdot \exp(t \cdot z_{k,max}), \end{cases} \tag{6}$$

where $\bar{x}_k$ is the center of $H_k$. Finally the system forward reachable set is a union of projected output polyhedral cells: $\bigcup\limits_{k=1}^N R_k^o$ where each cell $R_k^o$ is associated with a density bound $[\rho_{k,min}, \rho_{k,max}]$.

**System reachable set probability computation.** Given an initial state distribution, we want to figure out the probability distribution of the system forward reachable sets, as well as the probability for the states land into a query set (e.g., the query set could be the unsafe region).

For an arbitrary initial probability density function (whose support is bounded), we can apply RPM to partition its support into cells[4] as in the last section. Finally, we obtain the reachable sets with bounded state densities $\{(H_k, \rho_k^{min}, \rho_k^{max})\}_{k=1}^N$ and the probability bound in each cell is:

$$\begin{cases} P_k^{min} = \text{Vol}(H_k)\rho_k^{min}, \\ P_k^{max} = \text{Vol}(H_k)\rho_k^{max}, \end{cases} \tag{7}$$

where $\text{Vol}(\cdot)$ computes the volume for a polyhedron. By computing for all input cells $\{H_k\}_{k=1}^N$, we can derive the system forward reachable set and the corresponding probability bound as $\{(H_k, P_k^{min}, P_k^{max})\}_{k=1:N}$. The backward reachable set probability can be computed in a similar fashion, by checking the intersection between the query output region and the output cells derived by RPM, computing each intersection's probability range by its volume and density bound and finally aggregating the probability of all intersections. Detailed computation is shown in Appendix B. [5]

---

[4] We can always further divide those input cells to make the bound tighter/ more precise, while still guaranteeing the Neural Network on each cell can be seen as an affine transformation.

[5] The RPM method cannot handle systems with higher than 4 dimensional state space in our experiments. But the technique discussed in Sec. 3.2 can work with any set-based NN reachability tools with slight modifica-

| System | Dim. | Control |
|---|---|---|
| Van der Pol Oscillator (vdp) | 2 | - |
| Double integrator (dint) [42] | 2 | NN |
| Kraichnan-Orszag system (kop) [66] | 3 | - |
| Inverted pendulum (pend) [67] | 4 | LQR |
| Ground robot navigation (rpbot) | 4 | NN |
| FACTEST car tracking system (car) [68] | 5 | Tracking |
| Quadrotor control system (quad) [42] | 6 | NN |
| Adaptive cruise control system (acc) [4] | 7 | NN |
| F-16 Ground collision avoidance (gcas) [69] | 13 | Hybrid |
| 8-Car platoon system (toon) [70] | 16 | NN |

Table 1: Benchmarks from low dimension academic models to complex systems with handcrafted or NN controllers.

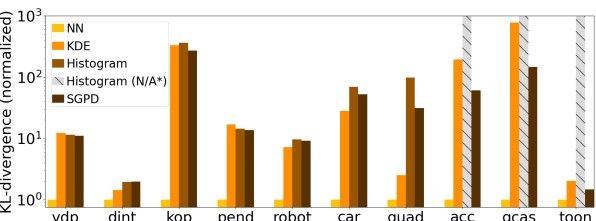

Figure 2: KL divergence with ODE-simulated density. Our method consistently outperforms other baselines. The histogram method cannot estimate the density for "acc", "gcas" and "toon".

## 4 Experimental evaluation in simulation

Here we show the benefits of learning *density concentration function* from Liouville PDE in state density and reachable set distribution estimation. More experiments are provided in Appendix C∼H.

### 4.1 Implementation details

**Datasets:** We investigate 10 benchmark dynamical systems as reported in Table 1. These benchmark systems range from low dimension academic models (2∼3 dimensions) to complex and even black-box systems (13∼16 dimensions) controlled by handcrafted or NN controllers. All controllers (except for the ground robot navigation example) are from the original references. More details (the model description and initial distributions) will be provided in Appendix C.

**Training:** For each system, we generate 10k trajectories through simulation with varied trajectory lengths from 10 to 100 time-steps, depending on different configurations of the simulation environment. We use 80% of the samples to train the NN as described in Sec. 3.1 and use the rest for validation. For the trajectory data, we collect the system states and compute for the system divergence term. For black-box systems, we use the gradient perturbation method to approximate the derivatives. We use feed-forward NN which has 3 hidden layers with 64 hidden units in each layer. We use PyTorch [71] to train the NN and the training takes 1∼2 hours on an RTX2080 Ti GPU.

### 4.2 Density estimation verification

We first test the density estimation accuracy of our learned NN. We compare our approach with other baselines including kernel density estimation (KDE), Sigmoidal Gaussian Process Density (SGPD) [72] and the histogram approach. For each simulation scenario, we first solve Eq. (2) to generate 20k ∼ 100k trajectories of (state, density) pairs, and treat this density value as the ground truth. For the KDE method, we choose an Epanechnikov kernel. We then measure the KL divergence between the density estimate of each method and the ground truth. As shown in Fig. 2, our approach has consistently outperformed KDE, SGPD and histogram approaches, with the largest reduction of 99.69% in KL divergence when compared with the histogram approach for the Kraichnan-Orszag system, while our method doesn't use any ODE generated density values during training. Also in high-dimension systems (dimension ≥ 7), the histogram approach fails to predict the density due to the curse of the dimensionality, whereas our approach can always predict the density, with a 30.13% to 99.87% decrease in KL divergence comparing to KDE. More plots will be given in Appendix D.

### 4.3 Reachable set distribution analysis

**Forward reachable set distribution analysis.** Being confident that our approach is able to provide an accurate state density estimation, we extend our NN to do distribution analysis, which is a valuable technique in safety-related applications like autonomous driving. Here we use an existing reachability tool RPM [21] to compute the forward reachable sets with probability bounds. Details about how to derive the density and probability bound for the reachable sets are presented in Sec. 3.2 and in Appendix B. Also, RPM was only able to parse the NN for Van der Pol, Double integrator, ground robot navigation, and the FACTEST car model. It fails in handling other high-dimension complex systems due to numerical issues when partitioning for the input set. Thus, we only report the results on those 4 models in this section. The main purpose is to show that for some systems the

---

tion based on the set presentation of the tool. Developing a better NN reachability tool that can scale to higher dimensional systems is another topic and is out of the scope of our paper.

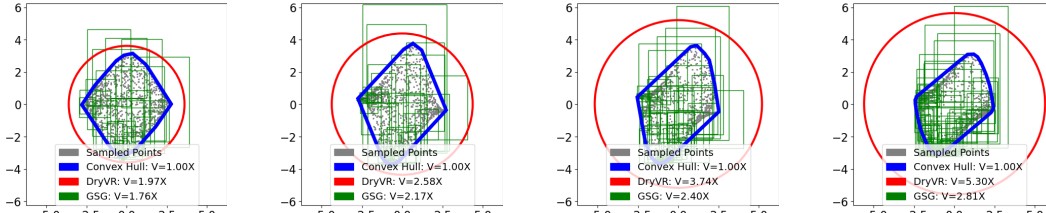

(a) The system forward reach set computed by the ConvexHull method(blue) [9], DryVR(red) [68] and GSC(green) [73]. We further show the relative volume of each kind of reachable set, comparing to the volume of the convex hull of the sampled points (gray). As time evolves, the conventional reachability methods often result in a larger over-approximation of the reachable sets with an increasing estimation error.

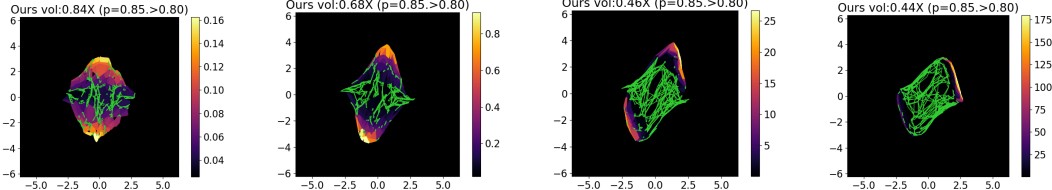

(b) The system forward reach set derived by RPM. The reachable sets are represented in polyhedral cells. The color ranged from dark purple to light yellow indicates the density inside the polyhedral cells. The edges colored in green indicate the boundaries of the RPM polyhedral cells with density below a threshold.

Figure 3: The Van der Pol oscillator forward reachable set comparison between (a) the worst-case reachability methods [68, 73] and (b) our probabilistic approach. Our approach clearly identifies reachable sets in high and low densities, and shows that as time evolves, the states will concentrate on a limit cycle, which is as expected.

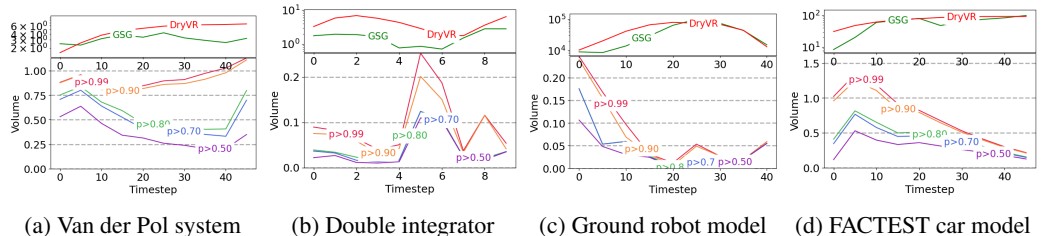

(a) Van der Pol system    (b) Double integrator    (c) Ground robot model    (d) FACTEST car model

Figure 4: The (relative) volume of reachable set with different probability level using our method, and comparison to other methods. The top part of each figure is in logarithm scale.

density tends to concentrate on certain states, where a small portion of the reachable sets contains the majority of states that are more likely to be reached. Therefore, our method can better quantify risks than worst-case reachability, by providing a flexible threshold for the probability of reachability.

We start with the Van der Pol oscillator. The initial states are uniformly sampled from a square region: $[-2.5, 2.5] \times [-2.5, 2.5]$. As illustrated in Fig. 1, the system states will gradually converge to a limit cycle. Using worst-case reachability analysis for this system will result in a very conservative over-approximation, and this over-approximation will propagate over time and lead to increasing conservativeness of the reachable set estimation. As shown in Fig. 3(a), for the worst-case methods like DryVR(red) [68, 74] and GSG(Green) [73], the volume of their estimated reachable set relative to the volume of the convex hull of the system states keeps increasing over time, from 1.9670X to 5.3027X for the DryVR [68] approach, and from 1.7636X to 2.8086X for GSG [73]. However, our method can give the probability bound for every reachable set in the state space as shown in the heatmap in Fig. 3(b), clearly identifying the region around the limit cycle in high density (bright color), and the rest space in low density (dark color).

We can also compute the relative volume of the reachable sets (comparing to the convex hull) preserving different levels of reachable probability, whose evolution over time reflects the system's tendency for concentration. As shown in Fig. 4, we use the above 4 systems and study the volume of the reachable set with probability threshold 0.50, 0.70, 0.80, 0.90, and 0.99. As expected, the relative volume will increase as the probability threshold increases. In all cases, there exist some time instances where a small volume of the reachable set actually preserves high probability, which shows

the state concentration exists in many existing systems. While as shown in Fig. 4, the worst-case reachability tools can only generate one curve which presents the (relative) volume of the reachable set that covers all possible states. Not surprisingly, the worst-case reachability tools give very conservative results. We believe using our proposed method to do reachable set distribution analysis will benefit future study for systems with uncertainty, and for systems where the failure case is inevitable but happens with a low probability. More comparisons with state-of-the-art reachability methods (Verisig [39], Sherlock [75] and ReachNN [40]) are shown in Appendix H.

**Online safety verification under different initial state distributions.** Since our approach learns the *density concentration function* instead of absolute density value, it has the flexibility to estimate online reach sets distribution with any possible (bounded support) initial distributions. Consider the safety verification for the ground robot navigation problem (shown in Fig. 5(a)): The probability of colliding with a obstacle in the center of the map is determined by the initial state distribution [6], which is parametrized as a truncated Gaussian distribution $N(\mu, \sigma^2 I)$ where $\mu$ and $\sigma$ measure the

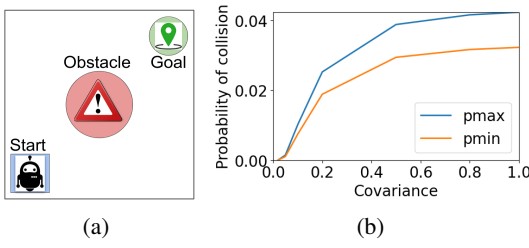

(a)  (b)

Figure 5: (a) A robot tries to reach the goal while avoiding the obstacle. (b) Probabilistic safety verification under different initial distributions. As the robot has less uncertainty in its initial position and velocity, it will have lower probability to collide.

expectation and uncertainty of the initial robot state. Our method can estimate the upper and lower bounds for the probability of colliding with the obstacle, and this safety evaluation process can run in faster than 50Hz with parallel computation and heuristic searching used (see details in Appendix E). As shown in Fig. 5(b), when the initial state uncertainty $\sigma$ decreases from $1.0$ to $0.02$, the upper and lower bounds for the probability of collision decrease to close to zero (from $0.04236$ to $3.5344 \times 10^{-06}$, where other worst-case reachability analysis methods can only report a collision is inevitable, without quantifying the corresponding risks. This also shows the advantage of our approach in adapting to different density conditions in computation without retraining or fine-tuning.

**Limitations and trade-offs of performing reachable set distribution analysis.** Experimental results show that our method can compute much less conservative probabilistic reachable sets than most worst-case reachability methods. This less conservative result benefits from RPM which can provide exact NN reachability analysis by sacrificing scalability. Therefore RPM also constrains us from performing online reachability analysis for high-dimensional systems or systems with large initial sets. Technically, we are solving a harder problem than worst-case reachability approximation, as we need not only the reachable set, but also the density over those reachable states. Like worst-case analysis, this is the reason why it can only scale to lower-dimensional systems and smaller initial sets when we want to perform accurate reachable computation. We believe that our approach can better quantify risks under different conditions, especially when unsafe is inevitable (similar to Fig. 5(a)). Our method can also give worst-case reachability by taking all output reachable cells produced by RPM regardless of their density. This worst-case reachability using RPM is less conservative than other NN reachability tools, at the cost of not scaling to high-dimensional systems.

## 5 Conclusion and discussion

In this paper, we propose a Neural Network (NN)-based probabilistic safety verification framework that can estimate state density, compute reachable sets and corresponding probability. Our Liouville-based NN can accurately estimate the state density even for high-dimension systems. Our probabilistic reachable set framework can handle nonlinear (and potentially black-box) systems with varying initial state distributions and can be used for fast online safety verification. We recognize that the task of computing probabilistic reachable sets is very useful, and our method is more helpful than worst-case reachability particularly when the system states are more likely to concentrate. One limitation of our approach is that the NN reachability tool we used (RPM) cannot handle high-dimension systems or cases where the initial set is very large, due to the numerical issues when partitioning for polyhedral cells. This limitation is due to the scalability and accuracy trade-off of NN reachability, which is an independent problem from our paper. We plan to explore other NN reachability methods and more complicated hybrid systems in real-world applications.

---

[6]How to compute the probability of a reachable set is discussed in Sec.3.2

**Acknowledgments**

The NASA University Leadership initiative (grant #80NSSC20M0163) and Ford Motor Company provided funds to assist the authors with their research, but this article solely reflects the opinions and conclusions of its authors and not any NASA or Ford entity.

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
