# OpenReview forum: "Learning Density Distribution of Reachable States for Autonomous Systems"
_robot-learning.org/CoRL/2021/Conference — CoRL2021 Poster_

### Official Review · Reviewer_mxEJ · 2021-07-07

**Originality:** Fair
**Technical Quality:** Good
**Clarity Of Presentation:** Good
**Impact:** 2

**Recommendation:**

Weak Accept: I recommend accepting the paper, but will not argue for my recommendation if the majority of other reviewers have a different opinion.

**Summary:**

This paper addresses the problem of estimating the density
distribution of reachable states of autonomous systems. The authors
train two neural networks (NNs): one to approximate the system
dynamics and one to approximate the density evolution over time. Once
the NNs are trained, a reachability tool, RPM, is used to provide
density estimates over sets of initial conditions and
densities. Evaluation is provided over 10 benchmarks selected from the
literature.

**Issues:**

I would like to see a discussion on the limitations of the proposed approach as compared with existing worst-case reachability methods. Furthermore, since neural networks are used to approximate the system dynamics and density, I am not sure how the reachability problem changes since it's not clear that the neural network can well approximate low-probability spaces.

**Reviewer Expertise:**

Excellent: Expert knowledge on the topic of the paper

**Strengths And Weaknesses:**

The idea of estimating density distributions, as opposed to standard
reachable sets, is interesting and relevant, especially in the context
of evaluating the risk associated with a certain task. The paper is
well written and the problem is clearly defined.

My main comment is that it is not clear how the proposed method
compares with state-of-the-art worst-case reachability tools such as
Verisig, Sherlock, ReachNN, etc. While I agree that estimating the
density distribution would be very useful, this problem is strictly
harder than estimating the reachable sets without taking into account
the probability distribution. In particular, the authors discuss
numeric issues as a challenge to their approach but I would argue that
this is not a numeric issue as much as it is an issue of whether to
include all reachable states, even those that have a very low
probability of being reached. On the other hand, if states with very
low probability are being ignored, then this needs to be clearly
stated in the problem statement and in the solution.

Related to the above, another interesting comparison would be with a
sampling-based method, which should perform well in low-dimensional
systems. In particular, given a sufficient number of simulations, one can
obtain a good estimate of the density distribution that can be
compared with the one produced in the paper.

**Summary Of Recommendation:**

I am recommending a weak reject because I think there is a mismatch between the author claims and the actual contributions. In particular, the authors seem to imply that the proposed method is more efficient than worst-case reachability methods ("...can be used for fast online safety verification"). I think the opposite is true, as discussed in my detailed comments -- this problem is strictly harder unless there are simplifying assumptions (which are not clearly stated).

---

> ### Comment · Reviewer_mxEJ · 2021-08-31
> **Post-Rebuttal Comment**
>
> I appreciated the rebuttal discussion and the effort the authors put in. My main concern had to do with explicitly outlining the limitations of the proposed method. That has been addressed, so I am changing my score to a weak accept.

---

### Official Review · Reviewer_bVFw · 2021-07-23

**Originality:** Very Good
**Technical Quality:** Very Good
**Clarity Of Presentation:** Good
**Impact:** 3

**Recommendation:**

Strong Accept: I recommend accepting the paper and will argue for my recommendation even if other reviewers hold a different opinion.

**Summary:**

This paper presents a data-driven method to approximate the probability density of reachable states for a nonlinear system. The proposed method learns the state density concentration function and the flow map (forward dynamics model that predicts the state at t given initial state x_0) using a neural network. The learned models are used to predict reachable states and corresponding density distribution.
The model is kind of a physics-informed NN, which is constructed upon the Liouville theorem. The proposed method formulates learning objectives (or apply constraints) based on the theorem that asserts that the state density follows the Liouville PDE (Eq. 1). The theorem characterizes the evolution of the state density along the trajectory.


The paper presents three experiments.
The training data is generated by simulating the benchmark systems. and evaluation is done by the density values obtained by solving the ODE (Eq. 2).

(1) The proposed method is evaluated 10 different systems including low-dimensional academic models such as Van der Pol oscillator and high-dimensional systems. In all examples, the proposed method could predict the state density better than the traditional KDE method in terms of the KL divergence to the ground truth.
(2) The proposed model is used to compute reachable sets of some of the benchmark systems. RPM is used to represent & compute reachable regions. In all the experiments, the proposed method could predict the forward reachable set close to the ground truth while the existing worst-case methods often fail in the systems where the states concentrate in small regions.
(3) Online safety verification experiment of the proposed method. Here the authors show that the reachability analysis can be done efficiently (20 ms per step) online using the learned model given arbitrary initial states.

**Issues:**

- In the first experiment, the ground truth labels are generated by solving the Liouville PDE. The fairness of this comparison is questionable. The proposed method is constructed to follow the Liouville PDE while the baselines (KDE & histogram) are not.  The experiment seems to be designed such that the proposed method outperforms others.
- It is unclear how the initial density is defined for each experiment.
- In the conclusion, there is a statement saying that the reachability analysis can be scaled up by taking other tools. I wonder If there is a more scalable method than the RPM that the authors are considering.
- Line 32, The author claims that "almost the entire space in the figure is reachable" if an existing reachability algorithm is used. This claim contradicts the result in Figure 2 (1). It is true for GSG or DryVR, but it seems like the Convex Hull method [9] does not exhibit such behavior.
- In Figure 2 and Figure 37-48 in supplementary, the gray points are not given in the legend. It seems like they are Monte Carlo samples that approximate the ground truth.
- The result & takeaway of the last experiment is not clearly presented. I had to look up the supplementary to understand the experiment.

**Reviewer Expertise:**

Fair: Some knowledge of the area

**Strengths And Weaknesses:**

- The problem definition is clear and well-presented. It was easy to understand what problem this work is addressing.
- The model is constructed such that the density concentration function (G) is independent of the initial state, which can be used to quickly compute the state density from any given initial density. This smart formulation enables the online reachability analysis shown in 4.3.
- The benefit of predicting the state density is that we can leverage different density conditions as demonstrated in supplementary E. However, this advantage is not clearly mentioned in the main text.
- The online reachability analysis has the potential for building a practical real-time control framework for safety-critical systems. Unfortunately, this paper does not provide a scalable solution for computing the reachability set.

**Summary Of Recommendation:**

I enjoyed reading this paper. The problem is well defined and the approach is very logical.
However, the experimental results and analyses are weak. The density values computed by solving Liouville PDE are used as ground truth to compare the proposed method against existing methods, which seems to be an unfair comparison because the proposed model is constructed to follow the Liouville PDE. The computation of the reachability sets presented in the last part is very interesting, but it is limited to low-dimensional systems.

---

### Official Review · Reviewer_HgHK · 2021-07-24

**Originality:** Good
**Technical Quality:** Good
**Clarity Of Presentation:** Very Good
**Impact:** 4

**Recommendation:**

Weak Accept: I recommend accepting the paper, but will not argue for my recommendation if the majority of other reviewers have a different opinion.

**Summary:**

In this paper the authors present a method for approximating the probabilistic forward evolution of an autonomous dynamical system by finding an approximate solution to the Liouville PDE. They train a fully connected neural network which learns the flowmap (i.e. trajectory) and the density function. The loss function is composed of two terms: prediction error for the dynamics, and PDE error. Moving from single-point estimation to set-based estimation, the authors additionally use RPM, a method which propagates the polyhedral partitions of the input space of a neural network as an affine transformation, in order to compute the forward reachable set and it's associated probability density. They showcase their algorithm on a variety of examples with improved KL divergence results than other methods.

**Issues:**

I would like these questions/issues to be addressed:
1.) Why are there two sets of parameters omega and theta different when only one single neural network is trained? Why not use two separate networks.
2.) There should be some more in-depth discussion of how the dynamics and dimensionality of the dynamical systems affect the number of samples required to closely approximate the Liouville PDE and the flow.

**Reviewer Expertise:**

Good: General knowledge of the area

**Strengths And Weaknesses:**

Strengths:
* It deals with an important problem in safety by analyzing probabilities rather than yes or no safety queries.
* The paper is clear and easy to read.
* They demonstrate the framework on a large variety of experiments.
* The results indicate that their approach obtains better KL performance than other methods.

Weaknesses:
* While RPM is a useful tool for verification, it is only used for a small number of examples in the paper given that polyhedral propagation does not scale well with input dimension. I would have preferred to see some of the information in the appendix replacing section 3.2.
* Some explanation of why they jointly model the flow and density would have been useful.


**Summary Of Recommendation:**

I think this paper proposes a method which is useful for safety applications. Taking a probabilistic perspective rather than a binary safe/unsafe approach can be of good use in robotics.

---

> ### Comment · Reviewer_HgHK · 2021-08-31
> **Thanks for the updates**
>
> Thanks for addressing our concerns. I like the paper and will keep my decision unchanged.

---

### Meta-Review · Area_Chair_J1gT · 2021-08-12

**Recommendation:** Accept (Poster)
**Confidence:** 4

**Metareview:**

### Final Meta-Review

After the revision and discussion process, reviewers agree that the authors have addressed their concerns to a satisfactory degree, both through clarifications during the discussion and through improvements to the paper's rigor and completeness. The final manuscript is significantly improved, and I can confidently recommend its inclusion in the CoRL technical program.


### Original Meta-Review

Reviewers agree that the paper presents interesting ideas around the use of density distributions to provide a richer risk quantification relative to the more commonly used reachable sets. However, they also express various concerns regarding the strength of the results and the accuracy of contribution claims.

In particular, Reviewer mxEJ points out a “mismatch between the author claims and the actual contributions”. This Area Chair agrees that statements about the better scalability and accuracy of state density estimation relative to worst-case (nondeterministic) analysis are not substantiated by the results. Reviewers HgHK and bVFw also express concerns related to scalability with state dimension not being clearly established. In addition, Reviewer bVFw points out multiple issues with clarity and rigor in the results and conclusions.

Adding to the above Reviewer concerns, the paper’s claim that this work is “the first to characterize and compute the reachability density/probability distribution for safety analysis” is incorrect as stated. There are in fact multiple works, in robotics alone, that compute probabilistic state distributions: for example, the work on confidence-aware predictions from Claire Tomlin and Anca Dragan’s groups, and on occupation measures from Ram Vasudevan’s group (also based on the Liouville equation).

The authors need to qualify the novelty of their contribution with respect to prior work, more rigorously state—and support—the advantages of their proposed approach, and adequately address the issues raised around the reported experimental results.

---

### Decision · Program_Chairs · 2021-09-13

**Decision:**

Accept (Poster)

**Comment:**

### Final Meta-Review

After the revision and discussion process, reviewers agree that the authors have addressed their concerns to a satisfactory degree, both through clarifications during the discussion and through improvements to the paper's rigor and completeness. The final manuscript is significantly improved, and I can confidently recommend its inclusion in the CoRL technical program.


### Original Meta-Review

Reviewers agree that the paper presents interesting ideas around the use of density distributions to provide a richer risk quantification relative to the more commonly used reachable sets. However, they also express various concerns regarding the strength of the results and the accuracy of contribution claims.

In particular, Reviewer mxEJ points out a “mismatch between the author claims and the actual contributions”. This Area Chair agrees that statements about the better scalability and accuracy of state density estimation relative to worst-case (nondeterministic) analysis are not substantiated by the results. Reviewers HgHK and bVFw also express concerns related to scalability with state dimension not being clearly established. In addition, Reviewer bVFw points out multiple issues with clarity and rigor in the results and conclusions.

Adding to the above Reviewer concerns, the paper’s claim that this work is “the first to characterize and compute the reachability density/probability distribution for safety analysis” is incorrect as stated. There are in fact multiple works, in robotics alone, that compute probabilistic state distributions: for example, the work on confidence-aware predictions from Claire Tomlin and Anca Dragan’s groups, and on occupation measures from Ram Vasudevan’s group (also based on the Liouville equation).

The authors need to qualify the novelty of their contribution with respect to prior work, more rigorously state—and support—the advantages of their proposed approach, and adequately address the issues raised around the reported experimental results.